Hydroclimatic variations reveal differences in carbon capture in two sympatric conifers in northern Mexico

González-Cásares Marcos 1 2
Pompa-García Marín 1
Venegas-González Alejandro alejandro.venegas@umayor.cl 3
Domínguez-Calleros Pedro 1
Hernández-Díaz José 4
Carrillo-Parra Artemio 4
González-Tagle Marco 5
1 Facultad de Ciencias Forestales, Universidad Juárez del Estado de Durango , Durango , México
2 Programa Institucional de Doctorado en Ciencias Agropecuarias y Forestales, Universidad Juárez del Estado de Durango , Durango , México
3 Hémera Centro de Observación de la Tierra, Escuela de Ingeniería Forestal, Facultad de Ciencias, Universidad Mayor , Santiago , Chile
4 Instituto de Silvicultura e Industria de la Madera, Universidad Juárez del Estado de Durango , Durango , México
5 Facultad de Ciencias Forestales, Universidad Autónoma de Nuevo León , Nuevo León , México
Marino Bruno
Electronic publication date: 2019 Jun 11
Publication date: 2019
Volume: 7
Electronic Location ID: e7085
Received 2018 Aug 13; Accepted 2019 May 2
Copyright: ©2019 González-Cásares et al.
Copyright year: 2019
Copyright holder: González-Cásares et al.
License: This is an open access article distributed under the terms of the Creative Commons Attribution License, which permits unrestricted use, distribution, reproduction and adaptation in any medium and for any purpose provided that it is properly attributed. For attribution, the original author(s), title, publication source (PeerJ) and either DOI or URL of the article must be cited.
License URL: https://creativecommons.org/licenses/by/4.0/

Keywords: Dendroecology, Climate change, Wood density, Aboveground biomass, Tree-ring analysis

Funding: CONACYT CB-2013/222522 LXVII Legislature of the State of Durango, México Universidad Juárez del Estado de Durango PIDCAF-UJED and DendroRed This research was supported by CONACYT (National Council of Science and Technology, project CB-2013/222522), the LXVII Legislature of the State of Durango, México, and the Universidad Juárez del Estado de Durango. The doctoral scholarship awarded to Marcos González-Cásares was supported by CONACYT, COCYTED (Council of Science and Technology of the State of Durango), and PIDCAF-UJED and DendroRed. The funders had no role in study design, data collection and analysis, decision to publish, or preparation of the manuscript.

==============================
Background

Forest ecosystems are considered among the largest terrestrial carbon sinks. The dynamics of forest carbon depend on where the carbon is stored and its responses to environmental factors, as well as the physiology of the trees. Thus, threatened forest regions with high biodiversity have great scientific importance, such as the Sierra Madre Occidental in Mexico. A comparative analysis of tree species can expand the knowledge of the carbon cycle dynamics and ecological processes in this region. Here, we examined the growth, wood density, and carbon accumulation of two threatened species (Pseudotsuga menziesii and Cupressus lusitanica) to evaluate their hydroclimatic responsiveness.

Methods

The temporal variations in the carbon accumulation patterns of two co-occurring species (P. menziesii and C. lusitanica) and their sensitivity to the local climate were studied using dendroecological techniques, X-ray densitometry, and allometric equations.

Results

The results show that the annual carbon accumulation in C. lusitanica is positively associated with the temperature during the current fall, while the carbon accumulation in P. menziesii is correlated with the rainfall during the winter of the previous year. The climatic responses are associated with the intra-annual variations of wood density and ring widths for each species. The ring width was strongly correlated with carbon accumulation in C. lusitanica, while the mean wood density was linked to carbon accumulation in P. menziesii.

Discussion

This study has implications for the carbon accumulation rates of both species, revealing differences in the carbon capture patterns in response to climatic variations. Although the species coexist, there are variation in the hydroclimatic sensitivity of the annual carbon sequestered by trunks of trees, which would be associated with tree-ring width and/or wood density, i.e., directly by anatomical features. The results are relevant to analyze the response to the variability of climatic conditions expected in the near future of the tree communities of Sierra Madre Occidental. Therefore, this study provides a basis for modeling the long-term carbon budget projections in terrestrial ecosystems in northern Mexico.

Introduction

Many of the world’s forest ecosystems are negatively responding to climate change, including changes in biodiversity, high mortality of tree communities, and increases in plagues and diseases (Allen et al., 2010). These phenomena are related mainly to the global increase in the intensity and severity of mega droughts in many regions, which have caused an acceleration in foliar senescence and forest decline in many tree species (Bigler et al., 2007; Sánchez-Salguero et al., 2010; Zeppel, Anderegg & Adams, 2012; Périé & De Blois, 2016; Venegas-González et al., 2018).

Tree growth in forest ecosystems is the most important terrestrial process associated with carbon dynamics. The total terrestrial carbon sink by forests is currently estimated at 2.4 ± 0.4 PgC per year (global net sink = 1.1 ± 0.8 PgC per year), being the temperate forests the second most important sink after tropical forest (Pan et al., 2011). However, the sequestration carbon by forest ecosystem depend on different factors, such as extreme weather events, land use change, stand age, forest disturbances, management practices and competition, among other ecological processes (IPCC, 2013). Thus, tree-ring growth dynamics contribute to the understanding of annual carbon uptake in forest ecosystems, allowing proposal of management actions for mitigation and adaptation to climate change (Babst et al., 2014a).

The Sierra Madre Occidental is the most extensive mountainous system in Mexico, and its forests represent the largest terrestrial oxygen- and biomass-producing ecosystems that remain in southwestern North America (González-Elizondo et al., 2012). The great biodiversity of this region allows for studies of concurrent species of great ecological importance, such as Cupressus lusitanica (Mill.), which grows in areas with high water availability, and Pseudotsuga menziesii (Mirb.) Franco, which represents a southern populations, with a limited distribution in Mexico.

Dendrochronology has been widely used in Mexico to study forest ecology and climate reconstructions (e.g., Acosta-Hernández, Pompa-García & Camarero, 2017). Some studies have been conducted with the intent to reconstruct precipitation (Cleaveland et al., 2003; Villanueva et al., 2011), teleconnections with El Niño–Southern Oscillation (Cleaveland et al., 2003), drought occurrence (Cerano et al., 2011), evaporation data (Pompa-García & Camarero, 2015), and analyze the seasonal climatic variations through early and latewood (Carlón-Allende et al., 2018). Tree-ring analyses (associated with other methods) have proven that the accumulation of carbon in different species (Pinus arizonica and Pinus cembroides) can be influenced by specific hydroclimatic conditions, site conditions, or the functional characteristics of each species (Pompa-García et al., 2018). However, this information is unknown in some threatened species of the Sierra Madre Occidental, such as Cupressus lusitanica and Pseudotsuga menziesii. As a result of their distributions along biogeographic gradients, these concurrent species can be used to evaluate the effects of limited water resources and the different conservation statuses on their growth dynamics, wood density, and carbon accumulation, as well as to assess the influence of hydroclimatic variability.

Several methods are used to estimate carbon content, such as total organic carbon (TOC; Houghton, 2005), X-ray densitometry (Taki, Nobori & Caceres, 2014; Pompa-García & Venegas-González, 2016), and allometric equations based on time series of tree diameters (Návar, 2009). These nondestructive methods can be used to analyze the temporal variations in carbon fixation of trees (Pompa-García et al., 2018).

In general, biomass evaluations assume that wood density is constant, ignoring the interannual variation caused by the climate and tree age (Babst et al., 2014b). The calculation of tree biomass can be obtained as a product of the volume and density of the wood. Current biometric studies rely on stem diameter growth estimated from tree rings (Babst et al., 2014b).

As tree growth responds differently to environmental conditions across tree species in Sierra Madre Occidental (Pompa-García et al., 2017a), there will also be variation in the total carbon accumulation by species. We used tree rings, allometric equations and wood density (WD) to examine the growth and carbon accumulation of C. lusitanica and P. menziesii trees located in the Sierra Madre Occidental and to evaluate their hydroclimatic response. We hypothesized that the variations in carbon accumulation reflect climatic conditions, according to the specific sensitivity of each species.

Materials and Methods

Area of study

The study was conducted in the Sierra Madre Occidental, in the state of Durango, northwestern Mexico (Fig. 1). This mountainous system of volcanic origin has an average elevation of 2,000 masl, reaching 3,000 masl in some areas, and it extends from the south of the Tropic of Cancer to the west of Durango, ending in southern Arizona (Aguirre-Díaz & Labarthe-Hernández, 2003).

Figure 1 Study site location and local climate diagram.

(A) The map shows the study area where Cupressus lusitanica and Pseudotsuga menziesii trees were sampled. (B, C) The climate diagrams illustrate the distribution of monthly precipitation (P) and mean temperature (T) for the time period 1946–2015 (El Salto meteorological station; located at coordinates 23°41′25″, −105°21′10″, at an altitude of 2,538 m).

This region has a temperate subhumid climate with a wet and cool summer due to the influence of the North American Monsoon (NAM). The region also has two characteristic dry seasons during spring and winter. The precipitation ranges from 1,000 to 1,200 mm annually. The main rainy season begins with the start of the NAM in late June or early July and ends in late September. Further low-intensity precipitation falls during the cold season (November–February). The maximum values of monthly temperatures occur from May to June; with minimum mean temperature of 7 °C, and a maximum mean temperature of 17 °C (CNA, 2016). The dominant soils in the area are cambisols, lithosols, regosols, and phaeozems (Aguirre-Díaz & Labarthe-Hernández, 2003), while in the study sites, we found mainly luvisol, regosol and cambisol.

Study species and sampling

We selected two coniferous species from Sierra Madre Occidental, Cupressus lusitanica and Pseudotsuga menziesii, due to their dendrochronological potential and importance for conservation and management (e.g., Pompa-García, Sánchez-Salguero & Camarero, 2017). C. lusitanica is located in humid sites while that P. menziesii is found in semihumid sites. Fifteen dominant individuals of each species were selected and sampled (diameter ≥10 cm at breast height (DBH), approximately 1.30 m above ground level). From each of these trees, two radial growth cores were collected at a height of 1.3 m with a nondestructive method using a Pressler borer (Ø = 5.1 mm). P. menziesii was collected at 2,747 masl and shows an average DBH and height of 36.3 cm and 18.1 m, respectively. C. lusitanica was collected at 2,651 masl and shows an average DBH and height of 31.5 cm and 16.4 m, respectively (Table 1). The field permit approval was granted by the Mexican Federal Government agency SEMARNAT (i.e., Secretariat of Environment and Natural Resources, No SGPA/DGVS 09456/16).

Annual density of wood

Radial wood cores were cut in the transverse direction, maintaining a thickness of 1.7 ± 0.2 mm, and these cores were kept in a room temperature (20 °C) and 50% relative humidity until they reached a stable moisture content of 12% (Tomazello et al., 2008). To determine the ring widths and wood densities, the samples were scanned from bark to pith using an X-ray densitometer QTRS01X Tree-Ring Scanner (Quintek Measurement Systems, Knoxville, TN, USA) at 0.08 mm intervals. The demarcation zone between the rings was automatically configured by the device and checked manually for each ring that was analyzed. For each year, the ring width (RW), mean wood density (MeanD), maximum wood density (MXD), and minimum wood density (MND) were determined. These four variables were used to analyze the response to local climate variability. The MeanD value was used to estimate the increase in aerial biomass and carbon.

Table 1 Characteristics of the species studied.

Period of time (TS), mean sensitivity (MS), expressed signal of the population (EPS), density (WD), carbon accumulation (C).

Species	Trees (Cores)	TS	MS*	EPS	RW	WD (Kg m−3)	C (Kg yr−1)	
C. lusitanica	15(12)	1855–2014	0.25	0.78	1.24 ± 0.35	462.3 ± 0.45(a)	2.2 ± 0.13(a)	
P. menziesii	14(13)	1901–2014	0.25	0.89	1.58 ± 0.33	550.5 ± 0.78(b)	3.7 ± 0.24(b)	
Notes.

Values are annual means ± SE. Means sharing a letter were not significantly different (p < 0.05) using a Mann–Whitney–Wilcoxon test. RW did not show significant differences.

Development of chronologies

We developed chronologies using the ring width obtained from the X-ray densitometer. Visual dating was then statistically validated by using the COFECHA program, which compares each series with a master chronology for each species (Holmes, 1983). For the construction of chronologies, the natural long-term growth trends (age and tree geometry) were eliminated using a negative exponential function to obtain standardized ring widths. Subsequently, an autoregressive model was applied to each of the standardized series to eliminate most of the temporal autocorrelation related to the growth of the previous year. Finally, a robust biweight average was used to obtain the chronologies of the residual indices for each species (mean = 1). This procedure was performed using the dendrochronology program library in R (dplR; Bunn, 2008) of the free statistical software R (R Development Core Team, 2015).

Aerial biomass and carbon estimation

The annual RW values were used to reconstruct the historical diameters of the trees and their increases in basal area. These values were then combined with the MeanD values to estimate biomass and, subsequently, the carbon allocated in that particular year. To estimate the biomass, we used the following allometric equation (1), which was proposed to estimate the carbon stocks in the forests of northwestern Mexico (Návar, 2009): (1) AWB=0.0752∗D2.4448∗2.0331p

where AWB = aerial biomass; D = normal diameter; and p = wood density. Different percentages of carbon concentrations in the biomass were used according to specific laboratory analyses (Yerena-Yamallel et al., 2012), as follows: C. lusitanica, 45.57%, and P. menziesii, 47.78% (Pompa-Garcia et al., 2017b).

Statistical analyses

Statistical analyses were performed using the Mann–Whitney–Wilcoxon test, with p < 0.05 considered significant in the evaluation of differences in radial growth, density, and carbon accumulation between species. This nonparametric test was used because the values of the variables did not comply with the basic assumptions of a normal distribution, according to a Shapiro–Wilks test (Zar, 2010).

The influences of climate (precipitation and temperature) on RW and WD were evaluated by analyzing the Pearson correlation coefficients, which were estimated for each species using residual chronologies. Autoregressive models were used to eliminate any temporal autocorrelation (Cook, 1985). Chronologies were compared with parameters of local climatic variability: Maximum temperature (Tmax), minimum temperature (Tmin), average temperature (Tmean), and precipitation (Pp). In addition, we evaluated the relationship between carbon accumulation and internal variables (RW and WD) and climatic influences (precipitation and temperature) by principal component analysis. Climatic data sets (1946–2014) were obtained from the “El Salto” meteorological station, located at coordinates 23°41′25″, −105°21′10″, which is less than 10 km from the study site at an altitude of 2,538 m (CNA, 2016).

Results

Growth, density and carbon accumulation

Table 1 shows the descriptive statistics of RW, density, and carbon accumulation for both tree species that were analyzed. The lengths of the chronologies, which included at least five trees, ranged from 113 (P. menziesii) to 159 (C. lusitanica) years, and the longest chronologies (with less than five trees) were those of C. lusitanica (244 years). The values of the expressed population signal (EPS) for P. menziesii were ≥0.85 over the entire period (estimated with at least five trees), mainly since 1920, showing that replication of the sampling was adequate. In the case of C. lusitanica, the usual EPS threshold of 0.85 was not reached (EPS = 0.78), but this does not mean that the samples were not adequately cross-dated. Rather, this means that the climate signal is lower in comparison to the sites where Pseudotsuga menziesii thrives (Wigley, Briffa & Jones, 1984). This sampling strategy has yielded good results for quantifying carbon accumulation through tree rings (Pompa-García et al., 2018). In this sense, before 1920, higher variability in the ring width series was observed (Fig. 2). The mean sensitivity was 0.25 for both species, showing that the trees react to the environment through their annual growth variability (Grissino-Mayer, 2001).

Figure 2 Tree-ring chronologies of the two study species.

Residual chronologies (black lines) and series (gray lines) of ring widths of the two study species for the best cross-dated period. (A) Cupressus lusitanica, (B) Pseudotsuga menziesii.

Both species had similar growth rates per year (1.24 mm for C. lusitanica and 1.58 mm for P. menziesii). However, P. menziesii had a higher mean wood density (550.5 kg m3 yr−1) (Table 1). In general, there was variation in the wood density measured since 1920, especially in P. menziesii (Fig. 3), where the three variables (maximum, minimum, and mean) exhibited negative trends. In contrast, C. lusitanica trees showed positive trends in mean wood density (MeanD) and minimum wood density (MND).

Figure 3 Mean values of wood density of the two study species.

Mean ± SE values of maximum density (MXD), minimum density (MND), and mean density (MeanD) of the two study species, for the period 1920–2014 (five trees per species). Positive (+) and negative (−) trends of density values are shown in right graphics. (A, B) Cupressus lusitanica, (C, D) Pseudotsuga menziesii.

Climatic influence on radial growth and wood density

The influences of climate variables on the RW and the WD variables (MND, MXD and MeanD) are shown in Figs. 4 and 5 for the period 1946–2014. Overall, P. menziesii showed more climatic sensitivity than C. lusitanica. For accumulated precipitation, the RW of P. menziesii trees was positively correlated with the rainfall during the winter or cold season (from October to April, r = 0.60, p < 0.001), while MND and MeanD exhibited negative correlations with the rainfall during winter and spring (from December to May, r <  − 0.36, p < 0.01). For C. lusitanica, the general pattern was that all the variables negatively reacted to the increase in rainfall during all months, with the change in MXD being significant throughout the year (r <  − 0.26, p < 0.05) (Fig. 4). Regarding the mean temperature, the RW and MXD of P. menziesii trees showed negative correlations with temperature during the previous and current growth years, which were significant for spring (March to May, r <  − 0.35, p < 0.01) and for annual mean temperature (r <  − 0.30, p < 0.05) in relation to RW, as well as for the temperature of the previous fall (September to November, r <  − 0.35, p < 0.01) related to MXD. For C. lusitanica, we observed a positive association between MND and MeanD and the mean temperature of the previous and current fall (r <  − 0.32, p < 0.01) (Fig. 5).

Figure 4 Relationships between residual chronologies (growth ring width (RW), maximum density (MXD), minimum density (MND), and mean density (MeanD) and accumulated precipitation).

The left and right columns represent the analysis by month and by season, respectively; that is, summer, fall, winter, spring, annual, and cold (October–April). Letters indicate growth periods (PY and lowercase = previous year, CY and uppercase = current year). Dashed horizontal lines indicate statistical significance at the 95% confidence level. (A) Cupressus lusitanica, (B) Pseudotsuga menziesii.

Figure 5 Relationships between residual chronologies (growth ring width (RW), maximum density (MXD), minimum density (MND) and mean density (MeanD)) and mean temperature.

The left and right columns represent analysis by month and by season, respectively; that is, summer, fall, winter, spring, annual, and cold (October–April). Letters indicate growth periods (PY and lowercase = previous year, CY and uppercase = current year). Dashed horizontal lines indicate statistical significance at the 95% confidence level. (A) Cupressus lusitanica, (B) Pseudotsuga menziesii.

Local and climatic influence in carbon accumulation

The annual carbon accumulation for 1920–2014 is represented in Fig. 6, which shows a higher carbon accumulation in P. menziesii trees (3.7 kg yr−1) than in C. lusitanica (2.2 kg yr−1) (Table 1). Since 1975, an increase in carbon accumulation in P. menziesii but a decrease in C. lusitanica (which has recovered since 2005) has been observed (Fig. 6A). This difference could be explained by the climatic influences (temperature and precipitation) and internal influences (radial growth and wood densities). These results seem to correspond to the high correlation between carbon and mean wood density of the first species (r = 0.48, p < 0.01) (Fig. 6B) and carbon and ring width in the second species (r = 0.45, p < 0.01) (Fig. 6C). Regarding climatic influences, we observed that carbon accumulation in C. lusitanica is linked to the temperature of the fall season (r = 0.40, p = 0.02), while carbon accumulation in P. menziesii is strongly associated with the precipitation during the previous winter (r = 0.58, p = 0.0002) (Fig. 7).

Figure 6 Temporal variation of carbon capture.

(A) Carbon accumulation per year in the two studied conifers. Gray lines indicate the standard error. (B) Relationship between carbon accumulation per year and ring width in Cupressus lusitanica. (C) Relationship between carbon accumulation per year and mean wood density in Pseudotsuga Menziesii. (D) Data trend of mean annual temperature and total precipitation per year.

Figure 7 Principal component analysis between carbon capture and internal influences (radial growth and wood density variables) and seasonal climatic influences (precipitation and temperature) for 1975–2014.

The yellow, green, blue and red circles indicate radial growth, the wood density variables, the precipitation variables, and the temperature variables, respectively. Win = winter, Fall = autumn, Spr = spring, Sum = summer. The dashed circle represents the maximum correlation with carbon accumulation. (A) Cupressus lusitanica (copheneticcorrelation = 0.79), (B) Pseudotsuga menziesii (cophenetic correlation = 0.81).

Discussion

The dendrochronological statistics were adequate for P. menziesii; however, C. lusitanica had a low EPS (0.78; Wigley, Briffa & Jones, 1984), which is consistent with previous studies carried out in the study area (Pompa-García et al., 2017a). This result is attributed to the fact that C. lusitanica is considered a species with low sensitivity to climatic variability because it inhabits areas near water bodies; however, it also responded to local climatic variability (Figs. 4, 5 and 7). The results indicate a strong relationship between the internal variables (RW and WD) and external factors (climatic variables) in the carbon accumulation capacity of the two species; this demonstrates the vulnerability of these ecosystems to global warming and environmental changes in northern Mexico.

The application of allometric equations associated with wood density and its response to climatic variability constitutes a reliable methodology that allows for the understanding of the variability in carbon accumulation under specific climatic conditions (Pompa-García & Venegas-González, 2016). This approach offers a temporal perspective of carbon accumulation and allows us to appreciate the intra-annual variation in woody biomass and consequently the carbon content (Pompa-García et al., 2018).

The inclusion of wood density as an indicator of climate variability improves the carbon capture estimates. Several studies carried out in the area have evaluated the responses to climatic conditions (González-Elizondo et al., 2005; González-Cásares, Pompa-García & Camarero, 2017; Pompa-García et al., 2017a), while others have used density for biometric purposes (González-Cásares, Yerena-Yamallel & Pompa-García, 2016; Pompa-García & Venegas-González, 2016). It has also been reported that the wood density of a species (Abies durangensis) that coexists with the species evaluated in the present study is more sensitive to temperature than the tree-ring width (González-Cásares, Pompa García & Venegas González, 2018). In this study, density was used to improve the estimation of carbon capture and its response to climate and thus facilitate a reliable and valuable comparative analysis to broaden the knowledge of the carbon accumulation in forest ecosystems.

The ring width of P. menziesii was positively correlated with the precipitation from the previous winter through March of the current year. Precipitation during the cold season positively affects tree growth because it recharges the soil water and triggers tree growth (Pompa-García & Venegas-González, 2016), while it also improves the photosynthetic activity of trees during the early growth season (Kerhoulas, Kolb & Koch, 2013). These results are consistent with those obtained for other conifers from nearby regions (Carlón-Allende et al., 2018; Cleaveland et al., 2003; González-Elizondo et al., 2005; Pompa-García & Camarero, 2015; Pompa-García & Venegas-González, 2016). The RW and MXD values of P.  menziesii showed a negative response to temperature, which is consistent with the results of another study in the region that reported a negative correlation between maximum temperature and WD in Pinus cooperi (Pompa-García & Venegas-González, 2016). This result could be attributed to the fact that the wood density of some conifers of the Sierra Madre Occidental is sensitive to high temperatures, which increases the cell wall thickness of the tracheids of latewood. In contrast, temperature has a positive influence on the MeanD and MND of C. lusitanica but not on the tree-ring width, which suggests that temperature controls the size of the cell (the lumen and the cell wall of the tracheids) but not the quantity (Thomas, Montagu & Conroy, 2007). In this species, it should be noted that the carbon accumulation is more associated with ring width than wood density (Fig. 7).

There is evidence that the increase in the global mean temperature is causing increased tree mortality due in water stress and attack by biotic agents (Allen, Breshears & McDowell, 2015). We observed a variation in carbon accumulation between the two species (Fig. 6A), that could be attributed to an increase in mean temperature and a decrease in rainfall in the region since 1980 (Fig. 6D, CNA, 2016). For P. menziesii, results showed that ring width and maximum density are negatively affected by temperature (Fig. 5). In addition, a high association with carbon accumulation was observed with mean wood density, which is strongly related to the rainfall during the cold season (Fig. 7A). All these findings would be affecting the growth of this species in northern Mexico. In western North America, an outstanding increase in mortality triggered by drought and high temperatures has been documented in P. menziesii, (Bentz et al., 2009).

In contrast for C. lusitanica, we observed a positive correlation with mean temperature that would positively affect carbon accumulation, especially during the fall (September–November) (Figs. 5A and 7B). A recent study found that tree growth of this species increased with temperature and did not seem respond to negatively to drought. Hence, it could be potentially favored with projections of global warming (Pompa-García, Sánchez-Salguero & Camarero, 2017). However, in other conifers from wet sites, if they are drought sensitivity and would be affected by the increase in temperatures by 2100 (González-Cásares, Pompa-García & Camarero, 2017; Pompa-García, Sánchez-Salguero & Camarero, 2017), so we believe that the response to climate varies between species.

As climate change progresses, there is evidence of global temperature increases being linked to an increasing frequency and duration of drought throughout much of the world (IPCC, 2013). The predicted climate changes are represented as an increase in the global average temperature, with an increase in aridity in some areas as well as in the frequency and severity of extreme droughts (Allen et al., 2010). In the projections of changes in vegetation caused by climate change, tree mortality becomes a central issue (Allen, Breshears & McDowell, 2015). When studying carbon capture in forests, it is important to have a general perspective on the effects of the climate on these ecosystems.

Combined with other methods, the calculation of biomass using growth rings provides more accurate estimations of carbon capture and improves the temporal resolution of periodic forest inventories (Babst et al., 2014c). The use of wood density for estimating biomass improves our understanding of the dynamics of carbon capture in these ecosystems (González-Cásares, Yerena-Yamallel & Pompa-García, 2016; Pompa-García & Venegas-González, 2016; Pompa-García et al., 2018). In general, the results show that low precipitation and high temperatures lead to substantial effect on carbon accumulation over the last 30 years. The increase in the mean annual temperature in the study area (CNA, 2016) that has taken place over the last years seems to have had a positive impact on carbon accumulation in some tree species, such as Cupressus lusitanica. Moreover, the negative trends of annual precipitation will cause reduced carbon uptake in some species, such as Pseudotsuga menziesii (Fig. 6D).

Conclusion

Cupressus lusitanica and Pseudotsuga menziesii exhibit different rates of carbon accumulation. This difference is attributed to the specific responses of each species to climatic conditions. C. lusitanica and P. menziesii showed different responses to seasonal climatic variability. In general, the carbon accumulation of P. menziesii responded significantly to the precipitation during the previous winter, while the temperature of the autumn season influenced carbon sequestration by C. lusitanica. In this sense, the negative and positive trends in precipitation and temperature in this region have caused negative and positive carbon accumulation trends in P. menziesii and C. lusitanica, respectively. The results of this study provide a basis for modeling the long-term carbon budget projections in terrestrial ecosystems in northern Mexico. Therefore, these results are of value for the evaluation of dynamic models of the global carbon balance.

Supplemental Information

Data S1 Raw measurements (ring width, wood density data, carbon chronologies and climatic data)

Click here for additional data file.

We thank the community known as “Ejido el Brillante” and to the forester responsible for the area (Dr. Javier Bretado), for the support provided for data collection. We thank the Dirección General de Vida Silvestre, SEMARNAT (Secretaría de Medio Ambiente y Recursos Naturales, Mexico), for facilitating field sampling. We thank Mario Tomazello Filho and Alci Albiero Junior for their support in preparation of samples and X-ray densitometry analysis. We also thank two other anonymous reviewers for their critical and useful comments.

Additional Information and Declarations

Competing Interests

Author Contributions

Field Study Permissions

Data Availability

The authors declare there are no competing interests.

Marcos González-Cásares conceived and designed the experiments, performed the experiments, analyzed the data, contributed reagents/materials/analysis tools, prepared figures and/or tables, authored or reviewed drafts of the paper, approved the final draft.

Marín Pompa-García conceived and designed the experiments, analyzed the data, contributed reagents/materials/analysis tools, prepared figures and/or tables, authored or reviewed drafts of the paper, approved the final draft.

Alejandro Venegas-González analyzed the data, contributed reagents/materials/analysis tools, prepared figures and/or tables, authored or reviewed drafts of the paper, approved the final draft.

Pedro Domínguez-Calleros, José Hernández-Díaz, Artemio Carrillo-Parra and Marco González-Tagle analyzed the data, authored or reviewed drafts of the paper, approved the final draft.

The following information was supplied relating to field study approvals (i.e., approving body and any reference numbers):

Field study permission was approved by the/15Secretaria Medio Ambiente y Recursos Naturales de Mexico (No SGPA/DGVS 09456/16).

The following information was supplied regarding data availability:

The raw measurements (ring width, wood density data and carbon chronologies) are available as a Supplemental File.

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
