# Peer review of "Hydroclimatic variations reveal differences in carbon capture in two sympatric conifers in northern Mexico"

_PeerJ, doi:10.7717/peerj.7085_

## Round 0.1 · original submission · Major Revisions

The reviewers have made suggestions and note concerns that when addressed will result in a much improved manuscript for publication in PeerJ.

For example:

1) The entire manuscript needs improvement in use of the English language.

2) Noting (1) above, the introduction lacks a clear thesis of what questions are addressed and how the manuscript is constructed.

3) “Authors fail to provide references on dendrochronological work conducted in Mexico which has identified climatic signals (temperature, rainfall) on tree growth, or in early vs. latewood. This is unfortunate because one of their studied species showed a pattern which is consistent with the findings of Carlón-Allende et al. 2016 & 2018. None of the references by Cerano or Villanueva are provided, which have also conducted analyses of climatic influences on tree growth for different conifer species in Mexico.”

4) “Authors should note that they did not measure carbon flows. They estimated carbon and biomass gain through growth in tree diameter, and through the use of one allometric equation. Hence, most of their argumentation about carbon fluxes is irrelevant to their study.”

Point 4 is particularly important as forest mensuration while appropriately employed in the study has limitations that should be addressed.

5) “There is little explanation on how the 15 individuals for each one of the two species were selected. Was there a criterion to collect cores from the largest and presumably oldest trees within forest stands? Where were the forest stands? Where all trees on south or north facing slopes?”

6) Validity of the findings:
“Presuming that trees were collected from sites where no site-effect might have modified their results, then the findings are credible. Because C. lusitanica was not very responsive to climatic variation, I am not convinced of the conclusions derived from this study about that species.”

Please address each of the reviewers comments and clearly note your changes in a separate document as well as annotating each change in a word document of the manscript.

We look forward to the revised manuscript and publishing this important study of carbon storage in Mexican conifers.

Reviewer 1 ·

Basic reporting

Under the consideration that I am not an English native speaker, the paper was clear for me

Experimental design

Methodology seems to me well done

Validity of the findings

Convincing results

Additional comments

Main comments
Convincing manuscript (I liked specially Figures 3, 4 and 5) that finds differences between Pseudotsuga menziesii and Cupressus lusitanica on the Mexican Sierra Madre Occidental, in sensitivity of dendrochronological variables in response to climate variables.
In my view, considering that the overall world-wide trend (due to climatic change) is increase mortality (e.g. Allen et al 2010) rather than a beneficial increase of growth of so, based on the findings of the authors, especially the strong sensitivity to winter rain for P. mensiezii , I wonder if the authors wanted to advance the hypothesis that expected increase of tree mortality (linked to climatic change) on Sierra Madre Occidental, might occur if happen one (or more) year(s) with a winter extremely dry and warm.
Conclusions need to be more explicit about what are those differences in climatic sensitivity between those two species (more details below).
There are minor changes needed indicated below, to improve clarity.
I suggest accepting it with minor changes.

Minor comments
Lines 55-56. “Such information enables the proposal of actions through which to understand the effects of climate change on forest ecosystems”. I believe that suggested actions, derived from gaining understanding of the processes, are for improving management to mitigate climatic change impacts. However, suggested actions are not for “understand” the effects of climate change. The understanding phase is previous to the action phase.
59-60. “when it is associated with density data”. Better: “when it is associated with Wood density data”.
77. “The Sierra Madre Occidental is recognized by…”. Better for an international audience: “The Sierra Madre Occidental along western México is recognized by…”.
83. “while the latter is located in areas close to water bodies”. ????? I doubt that. Well, not in my limited experience. Perhaps authors aimed to say “while the latter is located in temperate-cold moist areas”.
99-102 and 173-174. Figure 1 Caption. “El Salto meteorological station”. Indicate Lat., Long. and elevation of that station.
320-324. OK, if the overall trend (due to climatic change) is increase mortality rather than a beneficial increase of growth of so, and you found, specially for P. menziesii a strong sensitivity to winter rain, I wonder if you wanted to advance the hypothesis that expected increase of tree mortality (linked to climatic change) on Sierra Madre Occidental, might occur if happen one (or more) year(s) with a winter extremely dry and warm?
338-344. Conclusions could be more explcity. You says that “C. lusitanica and P. menziesii presented different responses to seasonal climatic variability”. OK, then, say what are those differences! P. menziesii is much more sensitive to previous winter rain (for example). “ and, as a consequence, the conservation and restoration of these species will require different and specific management measures” OK, I agree, but, what kind of specific measurements would you envisioned, given your results? Perhaps develop a bit of that on the Discussion section.

On Figures 4 and 5, I do not see the “Letters and arrows that indicate growth periods”. They are absent in my available version.

Reviewer 2 ·

Basic reporting

This is the weakest portion of the manuscript. Writing is not clear, and this obscures the relevance of the questions addressed and the importance of the results. I started making notes on the manuscript as I began to read it, but soon realized that I would have to correct most paragraphs to make it readable to an English speaking audience, so I capitulated! I seriously suggest that a professional English native speaker corrects the style throughout the manuscript.
Authors fail to provide references on dendrochronological work conducted in Mexico which has identified climatic signals (temperature, rainfall) on tree growth, or in early vs. latewood. This is unfortunate because one of their studied species showed a pattern which is consistent with the findings of Carlón-Allende et al. 2016 & 2018. None of the references by Cerano or Villanueva are provided, which have also conducted analyses of climatic influences on tree growth for different conifer species in Mexico.
Most sections of the manuscript are adequate, except for the introduction. It fails to provide enough evidence for the relevance of the questions addressed.

Experimental design

As stated above, research questions are not clearly justified in the introduction. Authors should note that they did not measure carbon flows. They estimated carbon and biomass gain through growth in tree diameter, and through the use of one allometric equation. Hence, most of their argumentation about carbon fluxes is irrelevant to their study.

Dendrochronological methods are adequate and clear. However, field and collection methods are not. There is little explanation on how the 15 individuals for each one of the two species were selected. Was there a criterion to collect cores from the largest and presumably oldest trees within forest stands? Where were the forest stands? Where all trees on south or north facing slopes?

Validity of the findings

Presuming that trees were collected from sites where no site-effect might have modified their results, then the findings are credible. Because C. lusitanica was not very responsive to climatic variation, I am not convinced of the conclusions derived from this study about that species.

---

## Round 0.2 · Major Revisions

The revised manuscript does not address the issues presented in the previous review. As you can see, Reviewer #2 states:

"Authors did not make substantial improvements to the manuscript. Writing is very confusing in several portions of the text. I took longer than I would have liked to in reviewing this manuscript because I indicated changes throughout. I hope that authors include my suggestions and re-elaborate entire paragraphs where they introduce concepts that are irrelevant to the main point of the study."

Please review the manuscript carefully in view of the comments above. We look forward to the revised manuscript.

Reviewer 1 ·

Basic reporting

English OK
New references were added, as requested by one reviewer.
Structure OK. Raw data available.
Discussion and conclusions are supported.

Experimental design

Methods follows usual methodology of the field

Validity of the findings

Discussion and conclusions are supported by their data.

Additional comments

I suggest to accept the ms.

Minor comments:
Comments on the tracked version.
L 95-97. “such as Cupressus lusitanica (Mill.) and Pseudotsuga menziesii (Mirb.) Franco. The first grows in areas with great water availability, while the second has a wider distribution. A comparative analysis could expand the knowledge.”.
I agree that Pseudotsuga menziesii has a very wide distribution across USA and Canada, but not in México. The sentence refers to Sierra Madre Occidental. Is needed to be more precise.

Comment for the pdf “clean” version
Figure 7 is absent. There is only the Figure caption.

If the authors made the above mentioned corrections, I do not need to see again the manuscript.

Reviewer 2 ·

Basic reporting

Authors did not make substantial improvements to the manuscript. Writing is very confusing in several portions of the text. I took longer than I would have liked to in reviewing this manuscript because I indicated changes throughout. I hope that authors include my suggestions and re-elaborate entire paragraphs where they introduce concepts that are irrelevant to the main point of the study.

Experimental design

Authors clarified most of my concerns in terms of experimental design. I wrote in the manuscript where further clarification is needed.

Validity of the findings

The main conclusion of the study, that is, that climatic variation can influence carbon uptake by trees, is well justified. Other side discussions, like the comparison of their method with other methods such as Eddy covariance, are unnecessary, unjustified, and only distract the reader from the main point of the study.

Additional comments

The few additions to the manuscript included in the second version did not address most of the limitations detected in the first version. Reviewing an article takes time, and it is done in the best of interests assuming that authors would reciprocate by improving the manuscript.

Annotated reviews are not available for download in order to protect the identity of reviewers who chose to remain anonymous.

---

## Round 0.3 · Minor Revisions

Thank you for the careful revisions and commentary. However, please respond to the final comments of reviewer #1. Please indicate your response and revision as you have done previously.

Reviewer 1 ·

Basic reporting

Basic reporting OK

Experimental design

Methodology used sound

Validity of the findings

Interesting findings that the two conifers have different sensitivity for different climate variables

Additional comments

One main point:
1) I wonder if there is no a mistake on the last paragraph of Discussion section (lines 350-360). They state that:
"The decrease in the mean annual temperature in the study area (CNA, 2016) that has taken place over the last 30 years seems to have had a positive impact on carbon accumulation in some tree species, such as Cupressus lusitanica. Moreover, the negative trends of annual precipitation will cause reduced carbon uptake in some species, such as Pseudotsuga menziesii (Figure 6d)."
Really there is a "decrease in the mean annual temperature" ?? With all the ongoing climatic change is that true? Your figure 6d, that you invoke in that paragraph, indicate an increase of temperature and a decrease of precipitation.
By the way, Figure 6 lacks a figure caption for panel (d).

Minor comment:
On several places of the revised version, scientific species names are not in italics.

---

## Round 0.4 · Minor Revisions

Thank you for the revised manuscript. Your article is nearly ready for publication in PeerJ.

Please note the final review comments to make your final revisions:

Only they missed again to provide a figure caption for Figure 6, panel (d). It could be something like:
"(d) Relationship between mean annual temperature and total precipitation per year ".
or
"(d) Data trend of mean annual temperature and total precipitation per year ".
or
"(d) Mean annual temperature and total precipitation regressed against year ".

Reviewer 1 ·

Basic reporting

New version improved

Experimental design

Methodology OK

Validity of the findings

Conclusions for each species supported

Additional comments

Authors corrected the wrong sentence in the last paragraph of the Discussion section.
I see that there are improvements on the English wording after the comments of the other reviewer.

Only they missed again to provide a figure caption for Figure 6, panel (d). It could be something like:
"(d) Relationship between mean annual temperature and total precipitation per year ".
or
"(d) Data trend of mean annual temperature and total precipitation per year ".
or
"(d) Mean annual temperature and total precipitation regressed against year ".

---

## Round 0.5 · accepted · Accept

Thank you for addressing the final comments by the reviewers. Your manuscript is now accepted for publication in PeerJ. Congratulations.